# Integrative GWAS and RNA-Seq analysis for target identification and virtual drug screening in colorectal cancer

Qinghui Liu[1,2]*, Yiyang Lei[1,2], Zixuan Liu[1], Jiale Han[1]

**1** Hebei Keqi Biotechnology Co., Ltd, Shijiazhuang, China, **2** Hebei Keqi Cancer Research Center, Shijiazhuang, China

* ffgh36@sina.com

## Abstract

### Background

Colorectal cancer (CRC) is a leading cause of global cancer-related mortality, necessitating the identification of novel therapeutic targets. Integrating genetic and transcriptomic data may reveal key molecular drivers of CRC progression and treatment opportunities.

### Methods

We performed a multiomics analysis combining genome-wide association study (GWAS) data (p < 1e-6) and RNA-seq data from the TCGA. Differential expression analysis (Limma) identified 24 consistently dysregulated genes (17 mRNAs, 7 lncRNAs) in CRC. Survival analysis was used to evaluate their prognostic impact on overall survival (OS), relapse-free survival (RFS), and post progression survival (PPS). Drug–gene interactions were explored via Enrichr, and virtual screening (PubChem) prioritized high-affinity compounds that target PYGL, a metabolic regulator.

### Results

Integration of GWAS and RNA-seq revealed that 24 CRC-associated genes, including *PYGL*, *SMAD7*, and *TCF7L2*, are involved in tumor metabolism and *Wnt*/*TCF* signaling. Survival analysis revealed that five genes (*CDKN2B*, *BOC*, *METRNL*, etc.) were significantly correlated with OS, RFS, and PPS. Ten small-molecule candidates targeting *PYGL* exhibited high binding affinity, suggesting their therapeutic potential.

### Conclusion

This study identified CRC-linked genes through GWASs and transcriptomics, highlighting their prognostic and druggable relevance. Computational drug repurposing pinpoints PYGL inhibitors as promising candidates, offering a translational framework for CRC therapy development.

**Data availability statement:** The gene expression data utilized in this study were obtained from the Genomic Data Commons (GDC) Data Portal (https://portal.gdc.cancer.gov/). Specifically, RNA-sequencing (RNA-seq) gene expression counts from the TCGA-COAD project were downloaded, with open access.

**Funding:** The author(s) received no specific funding for this work.

**Competing interests:** The authors have declared that no competing interests exist.

## Introduction

Colorectal cancer (CRC) accounts for approximately 10% of global cancer cases and is the second leading cause of cancer-related mortality [1,2]. In China, the annual incidence rate of CRC is 9.2%, ranking fourth among all cancers. In terms of both incidence and mortality, colorectal cancer rates were significantly higher in males than in females and greater in rural areas than in urban areas. The incidence increases significantly with age, especially after 40 or 45 years of age [3].

Single nucleotide polymorphisms (SNPs) are among the most common forms of genetic variation and involve single nucleotide alterations at specific genomic positions. These variations differ among individuals and can influence various phenotypic traits. SNPs can affect gene function, making them key determinants of susceptibility, progression, and prognosis in diseases such as CRC. The functional impact of SNPs depends on their genomic location and mutation type, with those occurring in coding or regulatory regions exerting more substantial effects on gene function. The TCGA database provides RNA-seq data across various cancer types, including gene expression profiles from tumor, adjacent, and normal tissues. Analyzing these datasets allows the identification of differentially expressed genes and their encoded proteins, many of which have emerged as key therapeutic targets in disease research, such as *SPAG5*, *MAGEA3* [4], and *TOP2A* [5]. Furthermore, a drug and disease prediction platform was developed on the basis of TCGA data [6,7].

However, single-method analyses of differential gene expression often have limitations, as they focus primarily on highly differentially expressed genes while potentially overlooking those with lower expression differences, thereby reducing the likelihood of identifying novel therapeutic targets or drug candidates [8]. To address this, the present study integrates GWAS and RNA-seq data to identify novel target genes and proteins for CRC treatment [9]. First, GWAS data from European and East Asian populations were analyzed to identify significant genetic variants associated with CRC. Next, RNA-seq analysis identified seven mRNAs and six long noncoding RNAs (lncRNAs) with significant differential expression in CRC patients. Finally, molecular docking analysis was performed to screen potential drug candidates targeting these differentially expressed genes. By integrating multiomics approaches, this study aims to identify novel biomarkers and potential therapeutic targets for CRC, providing a theoretical foundation for future precision medicine strategies.

## 1. Materials and methods

### 1.1 Study design

The study design process is illustrated in Fig 1.

### 1.2 Data acquisition

GWAS data related to CRC in European populations were obtained from the GWAS Catalog database. Datasets were filtered on the basis of case–control status and sample size, with smaller sample sizes (e.g., ncases < 5,000) excluded to ensure statistical robustness. The GCST90018808 dataset was selected (Table 1),

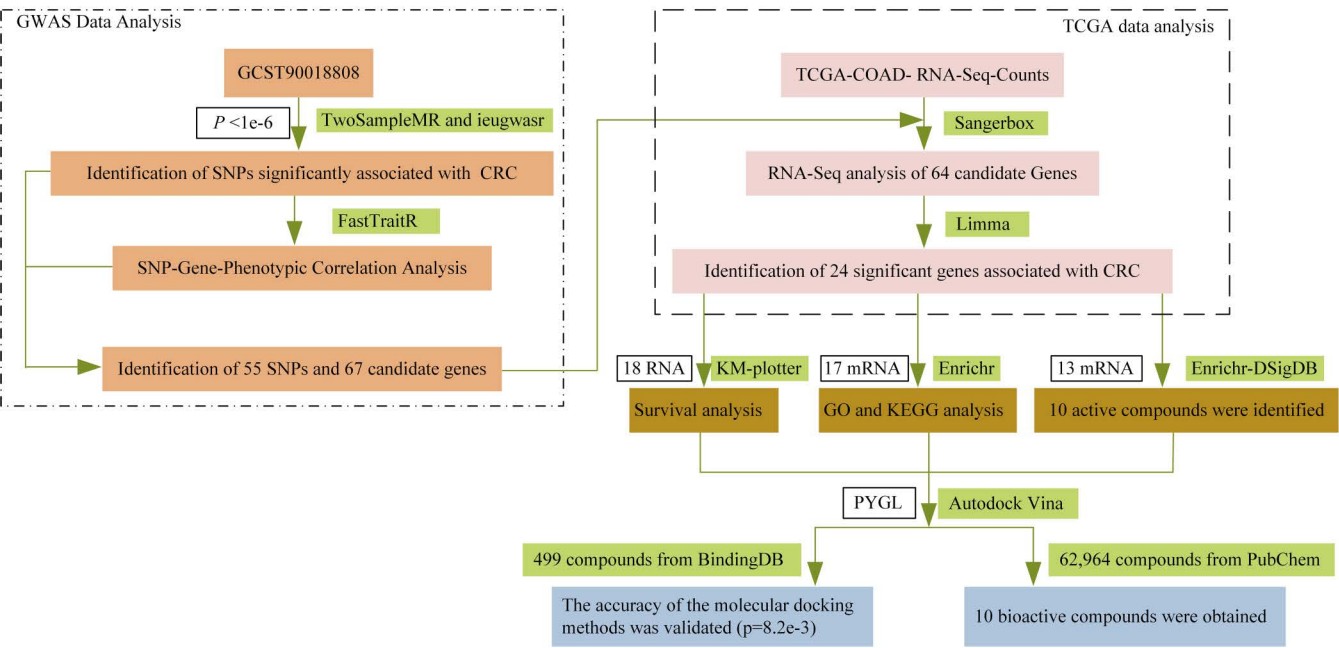

**Fig 1. Schematic representation of the study design.**

**Table 1. Summary of GWAS datasets for colorectal cancer analysis.**

| GWAS ID | Case (n) | Control (n) | Number of SNPs | Population | Ref. |
|---|---|---|---|---|---|
| GCST90018808 | 6,581 | 463,421 | 25,843,452 | European | [10] |
| | 8,305 | 159,386 | | East Asian | |

encompassing a total of 637,693 participants, including 14,886 CRC patients and 622,807 controls. The study population was exclusively European and East Asian, and both male and female participants were included to minimize allele frequency biases caused by population stratification and linkage disequilibrium (LD). Since this study utilized publicly available summary-level data, no additional ethical approval or informed consent was needed.

## 1.3 GWAS analysis

GWAS data analysis was conducted via the TwoSampleMR and ieugwasr packages in R (version 4.4.2). Chen et al. [11] employed a conditional p value threshold of $p < 1e-6$ to identify independent association signals and discovered several potential CRC susceptibility genes that had not been reported previously. SNPs with a significance threshold of $p < 1e-6$ were retained for further analysis, while weakly associated SNPs were filtered out. The functional annotation of the SNPs was performed via the FastTraitR package to assess the relationships among the SNPs, genes and phenotypes, facilitating the identification of CRC-associated genes. CMplot was used to construct a Manhattan plot, providing a visual representation of the positional distribution of significantly associated SNPs.

During linkage disequilibrium (LD)-based SNP screening, linked SNPs are typically removed, retaining only a single representative SNP. However, this selected SNP may have limited prior research or lack associated phenotypic data, potentially leading to the exclusion of SNPs with known phenotypic associations. To address this, we first conducted phenotypic analysis to identify SNPs with available phenotypic data, followed by LD analysis and subsequent investigation of the selected SNP loci.

LD-based instrumental variable selection was carried out via the ieugwasr package, with the parameters set to $r^2 < 0.1$ and a window size of 100,000 kb. To minimize bias from weak instrumental variables, SNPs with an F statistic < 10 were excluded [12].

## 1.4 RNA-seq differential expression analysis

RNA-seq data for colon cancer types were retrieved and analyzed via Sangerbox (http://sangerbox.com/tool.html) [13]. Gene identifiers (ENSG_ID) were converted to GeneSymbols. First, the expression matrix was obtained, and genes and samples with more than 50% missing (NA) values were removed. Next, missing values were imputed via the impute. knn function from the R package impute, with the number of neighbors set to 10,000 for data imputation, and data normalization was performed via $log^2(X + 1)$ transformation. The expression profiles of genes identified through GWAS were extracted, and differential expression analysis was conducted via the Limma package [14]. The analysis criteria included a fold-change threshold of 2.0 and a significance threshold of FDR < 0.01 (|logFC| > 1.0 and -log10(FDR) ≥ 2.0) [15]. Volcano plots were constructed to visualize the results, highlighting genes with significantly altered expression in CRC.

## 1.5 Survival analysis across all genes in colon cancer

Survival analysis was performed via the Kaplan–Meier plotter platform (https://kmplot.com/analysis/) [16]. Colon cancer was selected as the screening target, and either the gene symbols or Affy ids of the test genes were input into the designated fields. Patients were stratified via the "auto select best cutoff" option, which evaluates all possible cutoff values between the lower and upper quartiles and selects the optimal performance threshold as the cutoff value. The generated p value does not include correction for multiple hypothesis testing, all other parameters were maintained as default settings in the database. Separate survival analyses were conducted for OS, RFS, and PPS.

## 1.6 Enrichment analysis and functional annotation

Gene Ontology (GO) and Kyoto Encyclopedia of Genes and Genomes (KEGG) pathway analyses were conducted to examine biological process variations via the GO biological process 2025 library and the KEGG 2021 human database on the Enrichr platform (https://maayanlab.cloud/Enrichr/) [17]. The results were filtered to display the top 10 enriched GO terms and KEGG pathways, with a focus on the biological process category.

## 1.7 Online drug screening

The Enrichr database (https://maayanlab.cloud/Enrichr/), specifically the DSigDB dataset, which comprises 4,026 small-molecule compounds and 19,513 genes, was used for online drug screening [17]. Significantly differentially expressed genes in CRC were queried in Enrichr to identify candidate drugs and their mechanisms of action, facilitating the selection of potential therapeutic agents for CRC.

## 1.8 Virtual screening

Protein structures and interaction networks of the significantly differentially expressed genes were retrieved from STRING version 12.0 (https://cn.string-db.org/) [18]. Structural data were obtained from the AlphaFold Protein Structure Database and the Protein Data Bank (PDB), with priority given to PDB entries. The selection criteria included the scientific name of the source organism (*Homo sapiens*), the experimental method (X-ray diffraction or electron microscopy), and a refinement resolution of ≤2.5 Å. For proteins lacking small-molecule ligands, binding sites were predicted via computational tools. If ligands were present, the ligand-binding region was defined as the active site for drug screening.

The protein encoded by the *PYGL* gene (PDB ID: 3DDS) was selected as the receptor, with its original ligand, 26B (molecular weight: 505.60 g/mol), used for docking. The 3DDS structure was determined by X-ray diffraction at a resolution of 1.80 Å. Only protein structures with well-defined active sites were retained for further analysis.

This study retrieved PYGL-targeting compounds from the BindingDB database (https://www.bindingdb.org/rwd/bind/index.jsp) [19]. Compounds with IC50 values of zero or nonspecific numerical data were removed. After the elimination of structurally identical compounds, 499 unique small-molecule compounds were obtained, all of which had experimentally determined $IC_{50}$ values against PYGL.

A total of 231,187 small-molecule crystal structures were downloaded from the PubChem database. Lipinski's rule of five was applied to assess drug likeness [20] with the following criteria: molecular weight (200--500 Da), hydrogen bond donors (≤5), hydrogen bond acceptors (≤10), lipophilicity (LogP: 1--5), and rotatable bonds (1--10). After filtering, 62,964 small-molecule crystal structures were retained, hydrogenated, and optimized as ligands for virtual screening.

AutoDock Vina software was used for virtual screening [21], with protein crystal structures serving as receptors and the original ligand-binding pocket defined as the active site. Potential drug candidates were identified on the basis of binding energy and docking conformations. The molecular docking results were visualized and analyzed via PyMOL (version 3.1.0).

## 2. Results

### 2.1 Identification of SNPs significantly associated with phenotypes through GWAS data analysis

A significance threshold of $p < 1e-6$ was applied to filter out weakly associated SNPs from the CRC GWAS dataset derived from European and East Asian populations. Functional annotation was performed, and gene names were extracted from the MAPPED_GENE field across the datasets, resulting in the identification of 67 genes significantly associated with various phenotypes. These phenotypes include tumor-related traits, hematologic disorders, diabetes, and others. Most genes contained between 1 and 5 significant SNPs, although certain loci presented 10 or more. For example, the *SMAD7* gene (chromosome 18: 48,922,435–48,927,678 bp) has more than 10 associated SNPs, including rs11874392, rs12953717, and rs12956924. The *CDKN2B-AS1* locus (chromosome 9: 22,003,368–22,125,504 bp) contains more than 60 SNPs, all of which are strongly associated with gene activity.

LD-based instrumental variable selection was performed via the ieugwasr package. Integration of the GWAS datasets led to the identification of 55 SNPs and 67 candidate genes in total (S1 Table). Manhattan plots and QQ plot for the GWAS dataset were generated via the CMplot package. The results indicated that significant SNP loci were distributed across multiple chromosomes (Fig 2).

### 2.2 Identification of genes significantly associated with CRC via both GWAS and RNA-seq data analyses

RNA-seq data for colon adenocarcinoma (COAD) were retrieved from TCGA via Sangerbox. The data were normalized, with colon cancer patients serving as the comparison group and adjacent or normal tissues serving as the control group. Differential expression analysis was performed via the Limma package. To ensure data integrity, we removed genes with zero expression values in >50% of samples, yielding 64 genes associated with phenotype were identified.

By applying a fold-change (FC) threshold of 2.0 and a false discovery rate (FDR) threshold of 0.01, a total of 7 upregulated and 17 downregulated genes were identified (Table 2). A volcano plot was generated to visualize the RNA-seq analysis results for colon cancer, highlighting some genes that exhibited significant differential expression in both the GWAS and RNA-seq datasets for CRC. Notably, *TMEM220-AS1*, *TMEM238L*, *SOX6*, *SMAD7*, *TCF7L2*, and *PYGL* were significantly downregulated in colon cancer, whereas *LINC02257*, *PCAT1*, *CASC8*, and *POU5F1B* were significantly upregulated in colon cancer.

In addition, 7 lncRNAs, such as *CDKN2B-AS1*, *TMEM220*-AS1, *PCAT1*, and *CASC8,* were identified. Although these lncRNAs do not encode proteins, they play a regulatory role in gene expression through various molecular mechanisms.

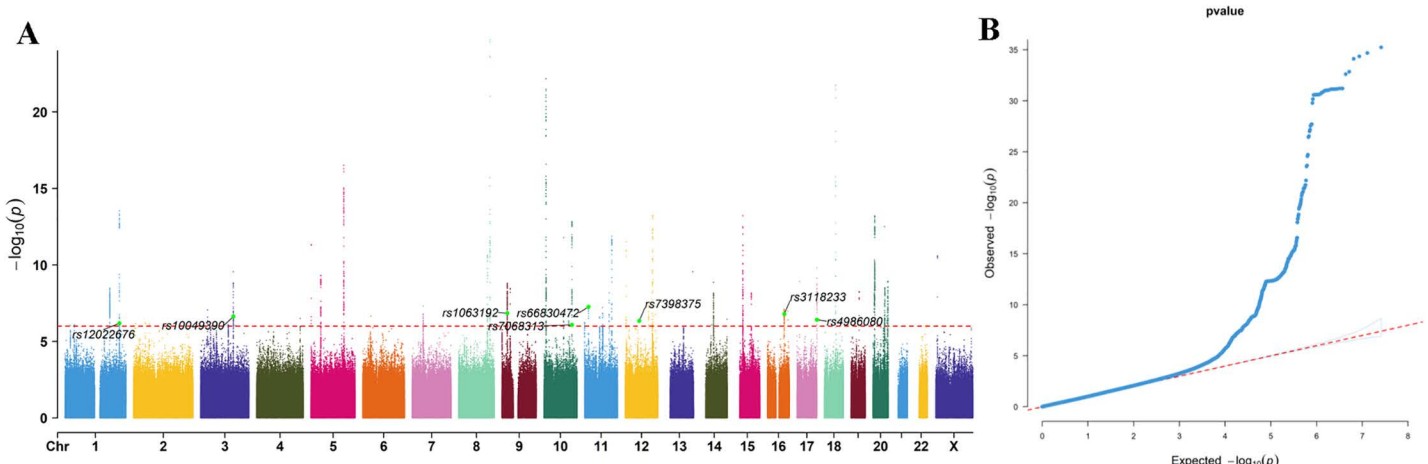

**Fig 2. Manhattan plot and QQ plot for GWAS ID: GCST90018808.** A: Manhattan plot; B: QQ plot. The significance threshold was set at 1e-6, with SNPs exhibiting significant associations with colorectal cancer (CRC) appropriately annotated. Multiple SNPs reached genome-wide significance ($p > 5e-8$), and their proximal genes exhibited significant differential expression in the RNA-seq data.

**Table 2. Information on selected SNPs and associated genes.**

| No. | GWAS | | | | Nearby gene | RNA type | RNA-seq | |
|-----|------|-----|-----|---------|-------------|----------|---------|---------|
| | SNP | Chr | Pos | p_value | | | logFC | P.Value |
| 1 | rs12022676 | 1 | 222146085 | **6.40E-07** | LINC02257 | lncRNA | 2.1226 | 7.80E-15 |
| | | | | | LINC02474 | lncRNA | 2.1521 | 2.96E-07 |
| 2 | rs10049390 | 3 | 133701119 | **2.33E-07** | SLCO2A1 | mRNA | −2.9041 | 1.41E-35 |
| 3 | rs139372065 | 3 | 28513403 | **4.41E-07** | ZCWPW2 | mRNA | −1.4994 | 3.99E-13 |
| 4 | rs72942485 | 3 | 112999560 | **8.07E-07** | BOC | mRNA | −2.3307 | 2.33E-16 |
| 5 | rs10505477 | 8 | 128407443 | 1.40E-33 | PCAT1 | lncRNA | 1.3592 | 1.57E-10 |
| | | | | | CASC8 | lncRNA | 1.9539 | 5.33E-12 |
| | | | | | POU5F1B | mRNA | 3.1255 | 2.44E-17 |
| 6 | rs1063192 | 9 | 22003367 | **1.43E-07** | CDKN2B | mRNA | −4.0373 | 7.20E-55 |
| | | | | | CDKN2B-AS1 | lncRNA | −6.6891 | 9.86E-71 |
| 7 | rs7068313 | 10 | 114725926 | **8.16E-07** | TCF7L2 | mRNA | −1.2813 | 1.19E-14 |
| 8 | rs61871279 | 10 | 101343705 | **8.80E-07** | NKX2–3 | mRNA | −3.5499 | 6.30E-44 |
| 9 | rs66830472 | 11 | 16007446 | **5.42E-08** | SOX6 | mRNA | −1.8222 | 2.38E-11 |
| 10 | rs7398375 | 12 | 57540848 | **4.47E-07** | LRP1 | mRNA | −1.1805 | 4.03E-08 |
| 11 | rs28611105 | 14 | 51359658 | 1.32E-08 | PYGL | mRNA | −1.1501 | 5.20E-06 |
| | | | | | ABHD12B | mRNA | −1.7702 | 0.0002864 |
| 12 | rs16969681 | 15 | 32993111 | 5.47E-09 | SCG5 | mRNA | −1.1292 | 2.77E-05 |
| 13 | rs2439411 | 15 | 66983982 | 7.37E-09 | LINC01169 | lncRNA | 1.8324 | 8.07E-08 |
| 14 | rs3118233 | 16 | 68733646 | **1.60E-07** | CDH3 | mRNA | 5.8195 | 1.12E-109 |
| 15 | rs1078643 | 17 | 10707241 | 1.26E-09 | TMEM220-AS1 | lncRNA | −3.4109 | 1.02E-43 |
| | | | | | TMEM238L | mRNA | −2.5739 | 5.84E-28 |
| 16 | rs4986080 | 17 | 81049741 | **3.80E-07** | METRNL | mRNA | −1.4039 | 1.58E-17 |
| 17 | rs7229639 | 18 | 46450976 | 1.74E-13 | SMAD7 | mRNA | −1.5507 | 8.96E-18 |
| 18 | rs6019378 | 20 | 47309716 | 3.73E-08 | PREX1 | mRNA | −1.1888 | 1.62E-10 |

*CDKN2B-AS1* and *TMEM220*-AS1 were significantly downregulated in colon cancer, whereas *LINC02257*, *PCAT1*, and *CASC8* were significantly upregulated in colorectal cancer (Fig 3). Meng et al. [22] developed mPEG-DSPE liposomes encapsulating siRNA-targeting lncRNAs, which effectively inhibited CRC progression.

## 2.3 Survival analysis

The Kaplan–Meier plotter background database is manually curated, incorporating gene expression data along with relapse-free and overall survival information sourced from GEO, EGA, and TCGA. In this study, survival analysis was performed on 18 screened genes (Table 3 and Fig 4). Among these genes, five genes, *CDKN2B*, *BOC*, *SCG5*, *PYGL*, and *METRNL,* demonstrated statistical significance (p value < 0.05) in terms of OS, RFS, and PPS, with hazard ratios (HRs) > 1, indicating that high expression of these genes increases mortality risk and reduces patient survival rates. These genes may represent promising targets for future pharmacological development. *SMAD7* expression was significantly different (p value < 0.05) across all three survival metrics, with an HR > 1 for OS and RFS and an HR < 1 for PPS, suggesting a complex relationship between its expression levels and colorectal cancer patient outcomes that warrants further investigation.

## 2.4 Enrichment analysis and functional annotation

The adjusted p values were calculated via the Benjamini–Hochberg method to correct for multiple hypothesis testing. All genes in the human genome were used as background. Only the top 10 significant results are presented in Table 4. GO enrichment analysis revealed that the *Wnt* signaling pathway, fat cell differentiation and regulation of epithelial-to-mesenchymal transition were the most highly enriched biological processes. The Wnt signaling pathway is a highly conserved signaling cascade that plays crucial roles in cell fate determination, tissue development, and tumorigenesis. In the canonical Wnt pathway, β-catenin degradation is inhibited, allowing it to accumulate and interact with TCF/LEF transcription factors (e.g., *TCF7L2*) to activate the transcription of downstream target genes.

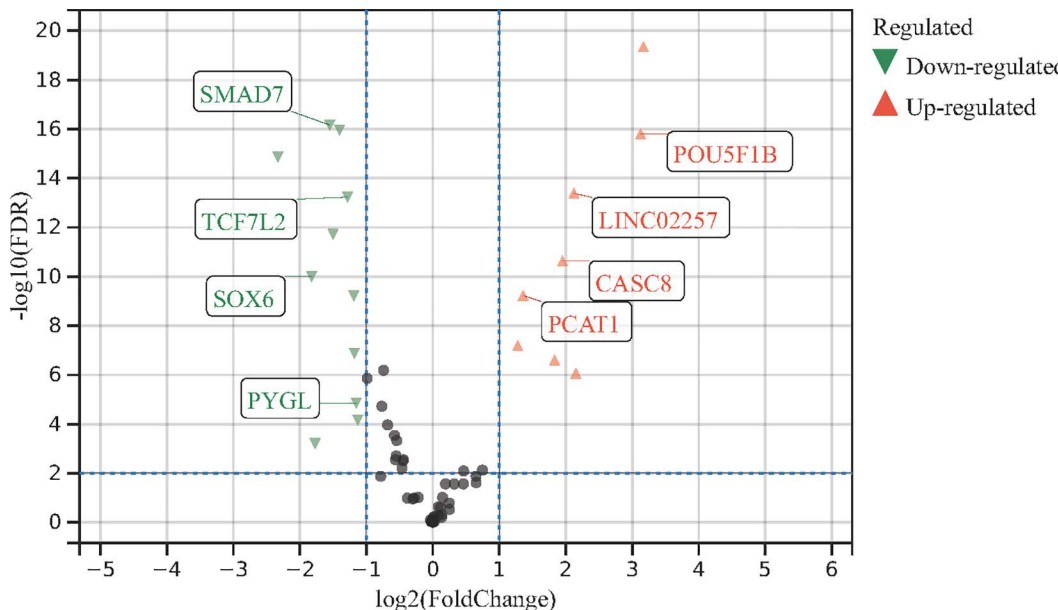

**Fig 3. Volcano plot of differentially expressed genes in colon cancer RNA-seq data.**

**Table 3. Genes whose expression is higher in CRC tumors and are correlated to OS, RFS, PPS.**

| No. | Gene symbol | Affy ID | OS | | | | RFS | | | | PPS | | | |
|---|---|---|---|---|---|---|---|---|---|---|---|---|---|---|
| | | | P value | FDR | HR | Cutoff value | P value | FDR | HR | Cutoff value | P value | FDR | HR | Cutoff value |
| 1 | **CDKN2B** | 236313_at | 2.2e-5 | 0.01 | 1.66 | 296 | 3.5e-5 | 0.01 | 1.72 | 225 | 0.0026 | 0.50 | 1.74 | 526 |
| 2 | **BOC** | 225990_at | 0.0003 | 0.03 | 1.54 | 257 | 9.9e-5 | 0.01 | 1.61 | 186 | 0.0008 | 0.10 | 1.76 | 262 |
| 3 | **SCG5** | 203889_at | 0.00056 | 0.05 | 1.43 | 212 | 0.0437 | 0.50 | 1.25 | 206 | 0.0045 | 0.20 | 1.54 | 274 |
| 4 | LRP1 | 200784_s_at | 0.0016 | 0.20 | 1.4 | 179 | 9.6e-9 | 0.01 | 2.29 | 88 | 0.1055 | 1.00 | 1.28 | 250 |
| 5 | ZCWPW2 | 243863_at | 0.0024 | 0.20 | 1.58 | 5 | 0.3117 | 1.00 | 1.13 | 14 | 0.2683 | 1.00 | 0.82 | 25 |
| 6 | **PYGL** | 202990_at | 0.0038 | 0.50 | 1.37 | 736 | 0.0003 | 0.03 | 1.67 | 312 | 0.0263 | 0.50 | 1.41 | 532 |
| 7 | PREX1 | 224925_at | 0.0248 | 0.50 | 1.33 | 223 | 0.0012 | 0.10 | 1.47 | 197 | 0.3142 | 1.00 | 1.19 | 205 |
| 8 | **SMAD7** | 204790_at | 0.0267 | 0.50 | 1.27 | 674 | 0.0002 | 0.02 | 1.52 | 658 | 0.017 | 0.50 | 0.68 | 423 |
| 9 | SLCO2A1 | 204368_at | 0.0292 | 0.50 | 1.27 | 358 | 0.0001 | 0.01 | 1.53 | 352 | 0.1291 | 1.00 | 0.79 | 405 |
| 10 | **METRNL** | 225955_at | 0.0433 | 0.50 | 1.28 | 840 | 1.3e-5 | 0.01 | 1.64 | 753 | 0.0095 | 0.50 | 1.58 | 759 |
| 11 | POU5F1B | 208286_x_at | 0.0453 | 0.50 | 1.24 | 296 | 0.3299 | 1.00 | 1.13 | 123 | 0.0852 | 1.00 | 0.77 | 205 |
| 12 | ABHD12B | 237974_at | 0.0612 | 1.00 | 1.26 | 93 | 0.0079 | 0.50 | 1.39 | 14 | 0.0672 | 1.00 | 1.37 | 69 |
| 13 | TCF7L2 | 212761_at | 0.1024 | 1.00 | 0.85 | 3825 | 1.8e-5 | 0.01 | 0.63 | 3786 | 0.0208 | 0.50 | 0.7 | 3841 |
| 14 | CDH3 | 203256_at | 0.1803 | 1.00 | 1.15 | 1042 | 0.0046 | 0.50 | 1.43 | 576 | 0.0752 | 1.00 | 1.35 | 1295 |
| 15 | SOX6 | 227498_at | 0.3515 | 1.00 | 0.89 | 60 | 0.0344 | 0.50 | 0.78 | 68 | 0.0271 | 0.50 | 0.66 | 130 |
| 16 | NKX2–3 | 1553808_a_at | 0.458 | 1.00 | 0.91 | 127 | 0.1049 | 1.00 | 0.83 | 137 | 0.0689 | 1.00 | 0.73 | 114 |
| 17 | CDKN2B-AS1 | 1559884_at | 0.002 | 0.20 | 1.46 | 4 | 0.0002 | 0.02 | 1.54 | 4 | 0.4442 | 1.00 | 1.14 | 6 |
| 18 | LINC01169 | 1563132_at | 0.0595 | 1.00 | 1.28 | 47 | 0.0219 | 0.50 | 0.76 | 8 | 0.0303 | 0.50 | 0.7 | 25 |

OS = Overall Survival, RFS = Relapse-Free Survival, PPS = Post-Progression Survival, HR = Hazard Rate, FDR = false discovery rate.

KEGG pathway analysis revealed several significant pathways, including the TGF-beta signaling pathway, gastric cancer and Cushing syndrome (Table 5), suggesting a potential role of these genes in cancer-related mechanisms and metabolic processes. The TGF-beta signaling pathway plays vital roles in regulating cell growth, differentiation, and immune function. The glucocorticoid receptor signaling pathway, which is implicated in Cushing syndrome, modulates a wide range of physiological processes, including glucose metabolism and immune responses. Starch and sucrose metabolism is central to maintaining energy homeostasis, regulating blood glucose levels, and supporting overall metabolic health. Notably, genes such as *TCF7L2*, *CDKN2B*, and *PYGL* are involved in multiple glucose metabolism-related pathways. We hypothesize that their downregulation in colorectal cancer tissues may disrupt glucose metabolism, thereby contributing to tumorigenesis.

## 2.5 Online drug screening for CRC-associated genes

The 13 identified genes were queried in the Enrichr database, and potential therapeutic compounds were screened via DSigDB (Table 6). These compounds exert their effects by targeting specific genes or their encoded proteins, thereby influencing key signaling pathways involved in CRC progression. As a result, these compounds and their derivatives hold potential as therapeutic agents for CRC treatment [23–25].

## 2.6 Virtual screening via molecular docking analysis of the receptor–ligand complex

A total of 17 mRNAs were identified, with their encoded protein structures available in the PDB or the AlphaFold protein structure database, among which only 3 possessed PDB IDs and contained small-molecule ligands. Based on the properties and size of the active sites, PYGL (PDB ID: 3DDS) was ultimately selected for further investigation [26], with the original ligand 26B serving as the docking ligand. Molecular docking was performed via AutoDock Vina, with the docking

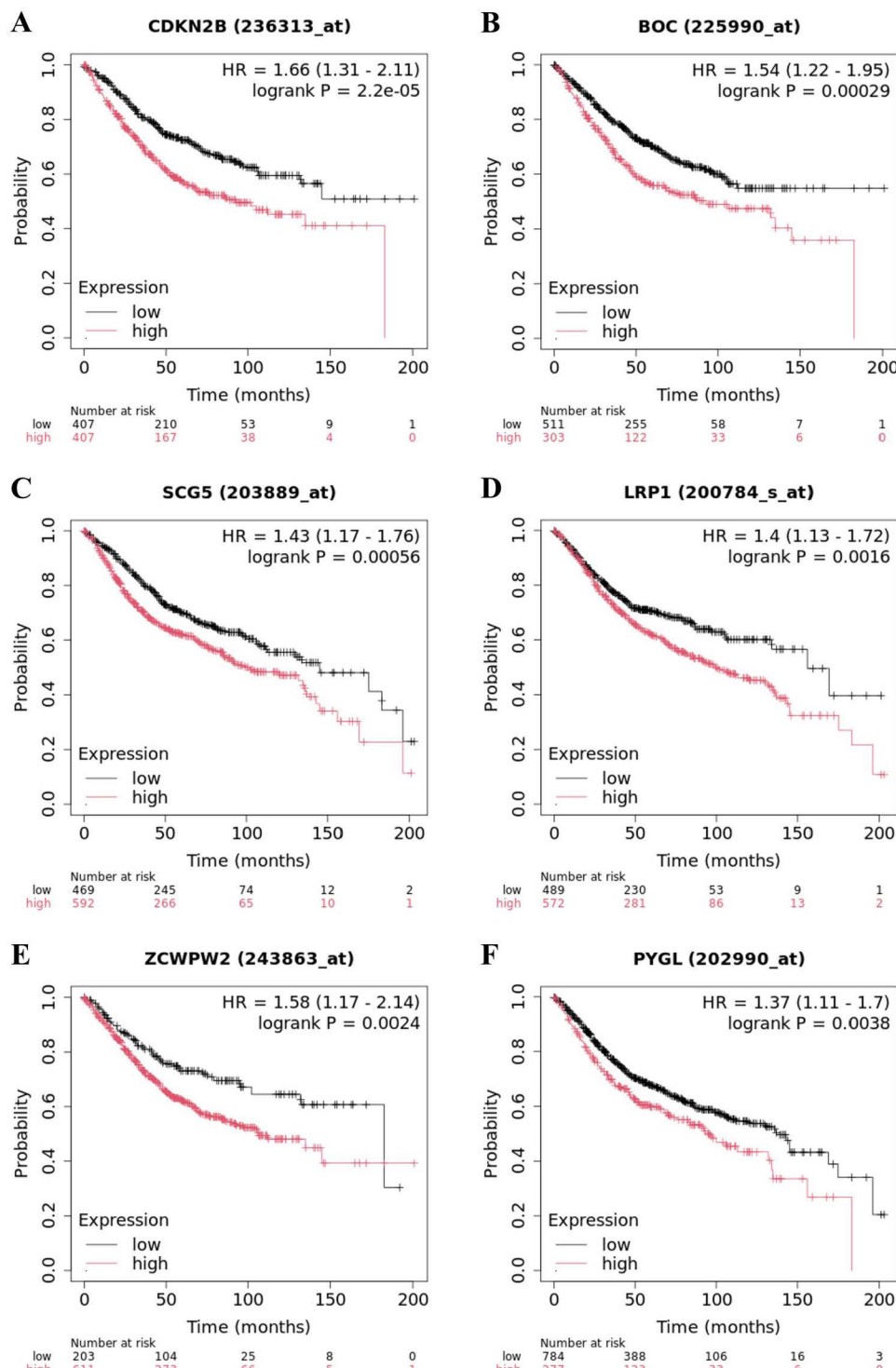

**Fig 4. Most significant druggable genes associated with Overall Survival.** HR = Hazard Rate, *CDKN2B*(A), *BOC*(B), *SCG5*(C), *LRP1*(D), *ZCWPW2*(E), *PYGL*(F).

**Table 4. Top 10 significantly enriched GO terms.**

| Term | Overlap | P-value | Adjusted P-value | Odds Ratio | Genes |
|---|---|---|---|---|---|
| Canonical Wnt Signaling Pathway (GO:0060070) | 2/63 | 8.67E-04 | 0.046307 | 54.43989 | TCF7L2; SMAD7 |
| Fat Cell Differentiation (GO:0045444) | 2/65 | 9.23E-04 | 0.046307 | 52.70635 | TCF7L2; METRNL |
| Positive Regulation of Cell Differentiation (GO:0045597) | 3/297 | 0.001045 | 0.046307 | 18.26716 | TCF7L2; BOC; SOX6 |
| Regulation of Epithelial to Mesenchymal Transition (GO:0010717) | 2/91 | 0.001798 | 0.046307 | 37.2603 | TCF7L2; SMAD7 |
| Glucose Homeostasis (GO:0042593) | 2/92 | 0.001837 | 0.046307 | 36.84444 | TCF7L2; PYGL |
| Regulation of Epithelial Cell Proliferation (GO:0050678) | 2/93 | 0.001877 | 0.046307 | 36.43773 | TCF7L2; CDKN2B |
| Cellular Response to Transforming Growth Factor Beta Stimulus (GO:0071560) | 2/98 | 0.002081 | 0.046307 | 34.53125 | SOX6; SMAD7 |
| Regulation of Transforming Growth Factor Beta Receptor Signaling Pathway (GO:0017015) | 2/120 | 0.003099 | 0.046307 | 28.06215 | CDKN2B; SMAD7 |
| Maintenance of DNA Repeat Elements (GO:0043570) | 1/5 | 0.003495 | 0.046307 | 384.2692 | TCF7L2 |
| Regulation of Ventricular Cardiac Muscle Cell Membrane Depolarization (GO:0060373) | 1/5 | 0.003495 | 0.046307 | 384.2692 | SMAD7 |

**Table 5. Top 10 significantly enriched KEGG pathways.**

| Term | Overlap | P-value | Adjusted P-value | Odds Ratio | Genes |
|---|---|---|---|---|---|
| TGF-beta signaling pathway | 2/94 | 7.70E-04 | 0.022317 | 61.79814 | CDKN2B; SMAD7 |
| Gastric cancer | 2/149 | 0.001918 | 0.022317 | 38.56948 | TCF7L2; CDKN2B |
| Cushing syndrome | 2/155 | 0.002073 | 0.022317 | 37.04575 | TCF7L2; CDKN2B |
| Hippo signaling pathway | 2/163 | 0.002289 | 0.022317 | 35.19077 | TCF7L2; SMAD7 |
| Kaposi sarcoma-associated herpesvirus infection | 2/193 | 0.00319 | 0.024879 | 29.61855 | TCF7L2; PREX1 |
| Starch and sucrose metabolism | 1/36 | 0.016087 | 0.081916 | 71.27143 | PYGL |
| Thyroid cancer | 1/37 | 0.01653 | 0.081916 | 69.28819 | TCF7L2 |
| Malaria | 1/50 | 0.022281 | 0.081916 | 50.87245 | LRP1 |
| Cholesterol metabolism | 1/50 | 0.022281 | 0.081916 | 50.87245 | LRP1 |
| Pathways in cancer | 2/531 | 0.022382 | 0.081916 | 10.51148 | TCF7L2; CDKN2B |

box centered at (x = 80.27, y = −97.68, z = 124.02). The calculated binding energy between the receptor and the original ligand was −12.3 kcal/mol, indicating a strong interaction. The docking conformation of the ligand exhibited a high degree of structural overlap with the original ligand (Fig 5). Both the crystal conformation and the docking conformation of the original ligand formed hydrogen bonds with key amino acid residues (Val40, Gln71, and Arg310), confirming the accuracy and reliability of the molecular docking method in predicting the optimal binding conformation of small-molecule compounds within the receptor's active site.

**Table 6. Potential therapeutic compounds identified through online screening.**

| Term | Overlap | P-value | Adjusted P-value | Odds Ratio | Genes |
|---|---|---|---|---|---|
| TERT-butyl hydroperoxide CTD 00007349 | 6/1341 | 1.69E-04 | 0.044751 | 10.47809 | CDKN2B;CDH3;LRP1;SCG5;PYGL;SMAD7 |
| Retinoic acid CTD 00006918 | 9/4258 | 6.22E-04 | 0.080978 | 6.666651 | POU5F1B;TCF7L2;PREX1;CDKN2B;CDH3;BOC;SLCO2A1;SOX6;SMAD7 |
| VALPROIC ACID CTD 00006977 | 12/8312 | 9.17E-04 | 0.080978 | 8.447711 | POU5F1B;TCF7L2;PREX1;CDKN2B;CDH3;BOC;SCG5;PYGL;METRNL;SOX6;ABHD12B;SMAD7 |
| indomethacin CTD 00006147 | 3/335 | 0.001478 | 0.09792 | 16.14513 | TCF7L2;SLCO2A1;SMAD7 |
| Rifampicin CTD 00006701 | 2/133 | 0.00379 | 0.194544 | 25.26081 | CDKN2B;SLCO2A1 |
| Decitabine CTD 00000750 | 5/1800 | 0.005882 | 0.194544 | 5.630145 | CDKN2B;CDH3;SLCO2A1;SCG5;SMAD7 |
| trichostatin A CTD 00000660 | 7/3584 | 0.006229 | 0.194544 | 4.587364 | PREX1;CDKN2B;CDH3;BOC;PYGL;METRNL;SMAD7 |
| Tesmilifene CTD 00001953 | 1/13 | 0.009064 | 0.194544 | 128.0385 | LRP1 |
| 4-tert-Butylphenol CTD 00000316 | 1/13 | 0.009064 | 0.194544 | 128.0385 | LRP1 |
| alsterpaullone MCF7 UP | 2/256 | 0.013419 | 0.194544 | 12.94751 | LRP1;SCG5 |

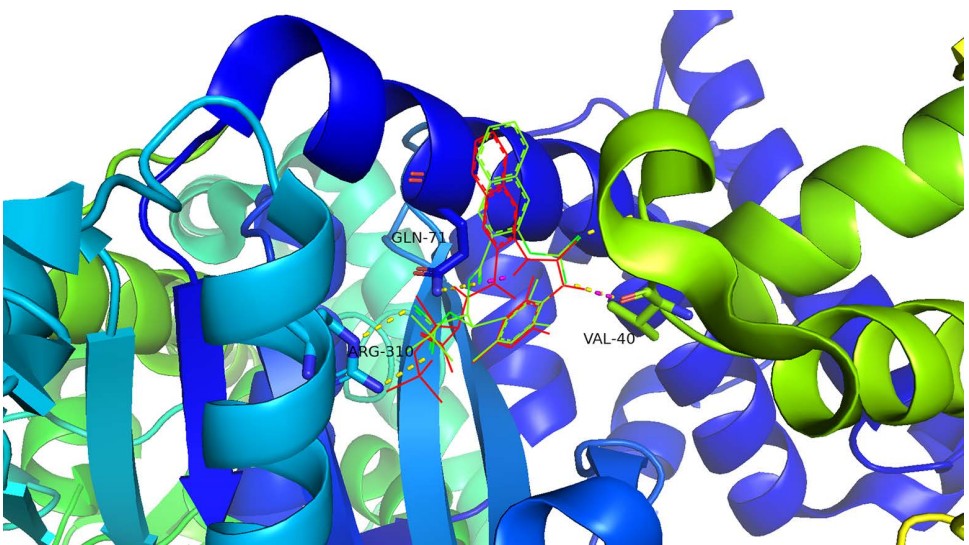

**Fig 5. Docking conformation of the original ligand at the active site.** The red structure represents the docked ligand conformation, with magenta bonds indicating hydrogen bonding interactions. The green structure denotes the crystal conformation of the protein-bound ligand, with yellow bonds highlighting hydrogen interactions.

A total of 499 small-molecule compounds associated with the *PYGL* gene were retrieved from the BindingDB database, with most exhibiting nanomolar-range $IC_{50}$ values against the PYGL protein. Molecular docking analysis via AutoDock Vina identified 155 compounds with binding energies ≤ −10 kcal/mol, whereas 497 compounds (99% of all analyzed compounds) had binding energies ≤ −7 kcal/mol. Only two compounds exhibited binding energies > −7 kcal/mol, two of which had molecular weights below 150 g/mol (S2 Table), suggesting that low molecular weights may have affected

docking accuracy. As shown in Fig 6, a positive correlation was observed between the binding energy and the Log(IC$_{50}$) value (p = 8.2e-3). These findings demonstrate that the screening platform based on AutoDock Vina can effectively identify potential active compounds that target the PYGL protease.

Small molecule compounds were retrieved from the PubChem database and filtered on the basis of Lipinski's Rule of Five to assess drug likeness. A total of 62,964 small-molecule compounds were selected, hydrogenated, and optimized for use as ligands in virtual screening. Using the PYGL protein structure (PDB: 3DDS) as the receptor and the original ligand binding site as the target, virtual screening was conducted with AutoDock Vina. This process identified 10 high-affinity compounds (Table 7), which are potential candidates for CRC treatment.

## 3. Discussion

This study provides a comprehensive molecular characterization of colorectal cancer (CRC) through integrated analysis of GWAS and transcriptomic data combined with computational drug discovery approaches. Our systematic investigation identified 24 CRC-associated genes, including key regulators such as *SMAD7* (a negative modulator of TGF-β signaling) [27,28], *TCF7L2* (a critical component of the *Wnt*/β-catenin pathway) [29], and *PYGL* (a metabolic enzyme involved in glycogenolysis) [30,31]. These findings not only reinforce established CRC pathways but also reveal novel molecular vulnerabilities, particularly in tumor metabolic reprogramming.

The computational drug screening pipeline identified ten high-affinity small molecules that target *PYGL*, suggesting potential for therapeutic repurposing. This metabolic target represents an innovative approach to disrupt cancer energy

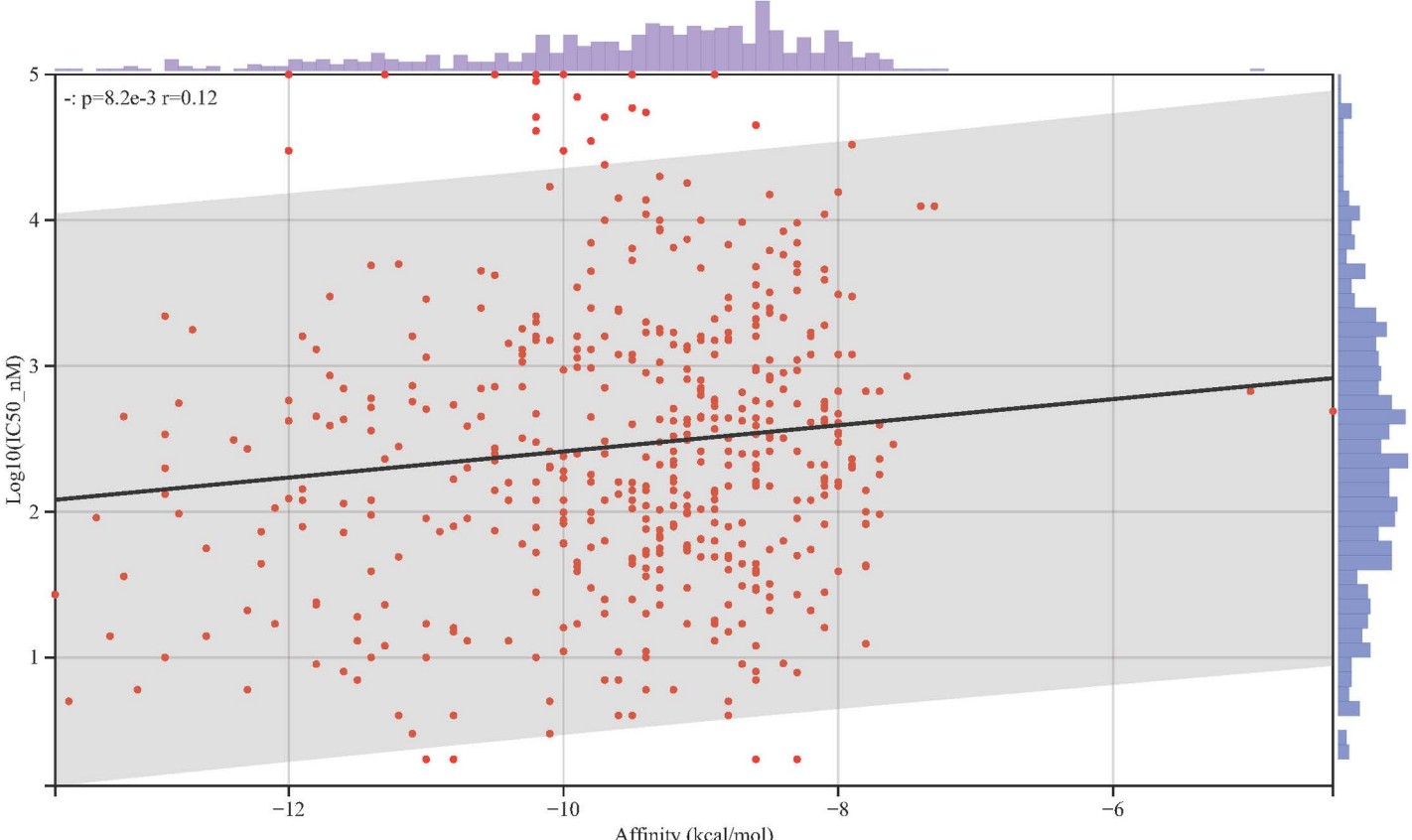

**Fig 6. Correlation analysis of the binding energies and IC$_{50}$ values of compounds retrieved from the BindingDB.**

**Table 7. Top 10 high-affinity compounds identified through virtual screening.**

| No. | Compound CID | Molecular formula | Molecular weight (g/mol) | Affinity (kcal/mol) |
|---|---|---|---|---|
| 1 | 139070664 | $C_{24}H_{21}N_3O_3$ | 399.4 | −11.3 |
| 2 | 44572643 | $C_{26}H_{27}N_2O$ | 383.5 | −11.0 |
| 3 | 139095568 | $C_{24}H_{27}NO_4$ | 393.5 | −11.0 |
| 4 | 139079778 | $C_{26}H_{24}N_2O_2$ | 396.5 | −10.9 |
| 5 | 15845721 | $C_{26}H_{22}N_4$ | 390.5 | −10.8 |
| 6 | 101542638 | $C_{24}H_{22}N_4$ | 366.5 | −10.5 |
| 7 | 139054227 | $C_{25}H_{25}N_3O$ | 383.5 | −10.5 |
| 8 | 139195689 | $C_{22}H_{24}N_4$ | 344.5 | −10.4 |
| 9 | 139116666 | $C_{21}H_{22}N_2O_3$ | 350.4 | −10.4 |
| 10 | 86237799 | $C_{23}H_{21}N_3O_2$ | 371.4 | −10.3 |

homeostasis, which is distinct from conventional kinase inhibitors. Furthermore, our analysis revealed that differentially expressed lncRNAs (e.g., *CDKN2B-AS1* and *TMEM220-AS1*) may function as epigenetic regulators in CRC pathogenesis, opening new avenues for RNA-targeted therapies [32].

While these computational predictions require experimental validation, the robustness of our findings is supported by the following:

1. Stringent statistical thresholds (p < 1e-6 in GWAS) [11].

2. Cross-platform consistency between the genomic and transcriptomic data.

3. Comprehensive molecular docking analyses.

4. Utilization of established drug–gene interaction databases.

Future directions should focus on the following:

1. Functional validation via CRISPR-based gene editing and organoid models [33,34].

2. Preclinical evaluation of *PYGL* inhibitors in relevant CRC models.

3. Development of LNP-encapsulated CRISPR systems for lncRNA modulation [35–37].

4. Investigation of combinatorial approaches targeting both metabolic and signaling pathways

This study establishes a framework for translating multiomics discoveries into therapeutic opportunities, highlighting the value of integrative computational biology in accelerating cancer drug discovery. The identified targets and compounds provide a foundation for developing next-generation CRC therapies that address the current limitations of targeted agents, particularly in overcoming drug resistance and improving treatment personalization.

## Conclusion

In summary, this study employed p < 1e-6 as the screening threshold for the colorectal cancer GWAS, combined with RNA-seq, GO, KEGG, and survival analyses. Survival analysis revealed that five genes (*CDKN2B*, *BOC*, *SCG5*, *PYGL*, and *METRNL*) were significantly correlated with overall survival (OS), relapse-free survival (RFS), and post-progression survival (PPS). Among these, *CDKN2B*, *BOC*, and *METRNL* showed p-values greater than 5e-8 in the colorectal cancer GWAS data. Subsequently, computer-aided drug design was utilized to screen multiple compounds exhibiting low binding energy with the PYGL protein, providing new therapeutic targets and directions for colorectal cancer treatment.

## Supporting information

**S1 Table. Phenotype-associated SNPs and genes identified in the GCST90018808 dataset.**
(XLSX)

**S2 Table. Binding energies and IC$_{50}$ values of compounds derived from BindingDB that target the PYGL protein.**
The R scripts used in the study are available via the following DOI: 10.5281/zenodo.15803098, v1.0.2.
(XLSX)

## Acknowledgments

We would like to thank Professor Baoen Shan for kindly proofreading the manuscript. We would also like to express gratitude to EditSprings (https://www.editsprings.cn) for the expert linguistic services provided.

## Author contributions

**Conceptualization:** Qinghui liu.

**Data curation:** Qinghui liu.

**Formal analysis:** Qinghui liu.

**Methodology:** Qinghui liu.

**Project administration:** Qinghui liu.

**Software:** Qinghui liu, Yiyang Lei, Zixuan Liu, Jiale Han.

**Validation:** Yiyang Lei, Jiale Han.

**Visualization:** Zixuan Liu.

**Writing – original draft:** Yiyang Lei, Zixuan Liu.

**Writing – review & editing:** Qinghui liu.

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
