## [Decision Letter · Decision Letter 0]

16 Jun 2025

Integrative GWAS and RNA-Seq Analysis for Target Identification and Virtual Drug Screening in Colorectal Cancer

PLOS ONE

Dear Dr. liu,

Thank you for submitting your manuscript to PLOS ONE. After careful consideration, we feel that it has merit but does not fully meet PLOS ONE’s publication criteria as it currently stands. Therefore, we invite you to submit a revised version of the manuscript that addresses the points raised during the review process.

We look forward to receiving your revised manuscript.

Kind regards,

Zhengrui Li

Academic Editor

PLOS ONE

Journal Requirements:

3. Thank you for uploading your study's underlying data set. Unfortunately, the repository you have noted in your Data Availability statement does not qualify as an acceptable data repository according to PLOS's standards.

Reviewers' comments:

Reviewer's Responses to Questions

**Comments to the Author**

1. Is the manuscript technically sound, and do the data support the conclusions?

Reviewer #1: Partly

Reviewer #2: Partly

Reviewer #3: Partly

Reviewer #4: Yes

Reviewer #5: Partly

Reviewer #6: Yes

2. Has the statistical analysis been performed appropriately and rigorously?

Reviewer #1: No

Reviewer #2: No

Reviewer #3: Yes

Reviewer #4: Yes

Reviewer #5: No

Reviewer #6: Yes

3. Have the authors made all data underlying the findings in their manuscript fully available?

Reviewer #1: Yes

Reviewer #2: Yes

Reviewer #3: Yes

Reviewer #4: Yes

Reviewer #5: Yes

Reviewer #6: Yes

4. Is the manuscript presented in an intelligible fashion and written in standard English?

Reviewer #1: Yes

Reviewer #2: Yes

Reviewer #3: No

Reviewer #4: Yes

Reviewer #5: Yes

Reviewer #6: Yes

Reviewer #1: Summary: This study attempts to find potential treatment targets and drug candidates for colorectal cancer by combining previously published genome-wide association studies (GWAS) with gene expression data (RNA-Seq) and drug screening through computational docking. The authors use existing public data to identify genes commonly linked to colorectal cancer in East Asian populations and suggest a few of them, such as PYGL and SMAD7, as potential targets. They then use molecular docking to screen for drugs that might bind to these targets. The results suggest a list of small molecules that could theoretically be repurposed to treat colorectal cancer. However, all analyses were performed with previously published datasets, and there was no experimental validation.

Strengths:

1. The manuscript is clearly written and well-structured, though excessively long for the novelty it presents.

2. Figures and tables are presented with clarity, though mostly re-confirm well-established findings.

Major Comments:

1. Lack of Novelty in Data or Methodology: The manuscript relies entirely on publicly available GWAS (e.g., bbj-a-76, GCST009869) and RNA-Seq (TCGA-COAD, TCGA-READ) datasets. No new data generation or significant methodological innovation is evident.

2. Absence of Experimental Validation

3. Overstated Claims: The manuscript claims to offer a "theoretical framework for drug repurposing strategies" and makes repeated references to “promising therapeutic targets,” yet all results are entirely in silico, and the novelty is limited.

4. Redundancy With Existing Literature: The identified targets (e.g., SMAD7, TCF7L2, PYGL) have been widely studied in colorectal cancer, and their roles are well-documented. The manuscript does not provide fresh insights into these genes functions or offer a new angle on their clinical relevance.

Reviewer #2: This manuscript presents results from a workflow that integrates GWAS and RNA-seq analyses for identifying risk genes in colorectal cancer (CRC) and conduct targeted virtual drug screening. The authors report on 13 potential risk genes, with the PYGL representing the most promising therapeutic target, for which ten small molecule inhibitors were identified as potential drug targets for CRC.

The following are my major comments:

(1) A significant portion of the Introduction discusses Mendelian Randomization (MR), a method for investigating the causal relationships between exposures and some outcome, typically disease phenotypes. And further, the analysis of archived GWAS data was done in part using the R package TwoSampleMR, with the aim of minimizing weak instrumental variables, a key step in any MR analysis. And yet, curiously, no MR analysis was actually performed in this study. This needs clarification.

(2) For the GWAS data analysis, the summary level data from three GWAS datasets were examined. Simply, SNPs with P-values less than 1E-5 and r^2 less than 0.001 within a large 10,000 kb (i.e., 10 Mb) sliding window were retained. Presumably there is a substantial overlap in the genome-wide SNPs tested in these datasets. If so, most of the SNPs are likely to have multiple association P-values (i.e., from each of three GWAS datasets). How was this handled? In the Results, a Manhattan plot is presented (Figure 2), but for only one of the three GWAS datasets (bbj-1-107). Further, the SNPs that passed the filtering thresholds are discussed per dataset, giving the reader the impression that each GWAS dataset was indeed examined examined separately, with each batch of SNPs simply lumped together to comprise the final list. If so, this is problematic. Typically some sort of meta-analysis is performed to combine evidence across the three datasets (as opposed to disregarding data and cherry-picking the best result per SNP). Common methods include inverse-variance weighted fixed-effects meta-analysis or random-effects meta-analysis (e.g., METAL, GWAMA). This will yield a single P-value and effect estimate per SNP across the three datasets. Please discuss.

(3) From the GWAS analysis, 27 genes "significantly associated with CRC" were identified (NOTE: in Figure 1, this is listed as 28 genes). Of these, 13 genes (including lncRNAs) show significant differential expression in colon cancer tissue (NOTE: gene expression results for rectal cancer (READ) are not provided, despite being analyzed separately). Although each of these 13 genes harbor at least one significant GWAS SNP related to CRC, the relationship of these SNPs to cis gene expression was not examined (i.e., eQTLs). SNPs can have disease risk effects that stem from deleterious changes to an encoded protein impacting function (e.g., nonsynonymous mutations), with *no* effect on gene expression levels. Thus, going from a SNP association with CRC to differential gene expression in CRC without actually examining the relationship between the SNP and gene expression to link the two appears somewhat problematic. Please discuss.

(4) For the enrichment analysis for GO terms and KEGG pathways, it was performed on a list of only 13 genes. Except in cases of exceptional enrichment, this list is too small to yield anything informative. Looking at the top results (Tables 3 and 4), most of the GO terms and pathways involve only a single gene in the list (with a few having two genes). This doesn't provide any real insight. The point of such tests is to find wider patterns that connect various genes in their function and/or biological processes and represent an overrepresentation of such characteristics. Finding a single gene from the 13 genes that is also among the genes involved in, say, "starch and sucrose metabolism" doesn't tell you much, despite an adjusted P-value less than 0.1. This type of gene annotation can be achieved using various databases without resorting to enrichment testing. The authors may want to reconsider how to conduct this step in their study design.

(5) The gene PYGL was selected for "virtual screening via molecular docking analysis". Why this gene was selected from the list of 13 genes is not clear to this reviewer. Please explain.

Reviewer #3: This manuscript integrates genome-wide association study (GWAS) data from East Asian populations with RNA-Seq data from TCGA to identify colorectal cancer (CRC)-related genes, followed by virtual drug screening using molecular docking. The study highlights several candidate targets including PYGL, SMAD7, and TCF7L2, and proposes ten small-molecule compounds with potential therapeutic effects.

While the concept of integrating multi-omics data for drug discovery is promising and timely, the manuscript has several methodological and interpretational weaknesses that need to be addressed before it can be considered for publication.

Major Concerns

Despite the mention of Mendelian Randomization and LD-based SNP filtering, there is no actual demonstration of SNP-gene expression links through eQTL analysis. Without this, the integration between GWAS and RNA-Seq remains superficial. eQTL or TWAS analyses are essential to strengthen the causal inference between SNPs and gene expression in CRC.

Key genes (e.g., PYGL, SMAD7, TCF7L2) are listed, but their roles in CRC progression are not deeply discussed. More mechanistic insight or network analysis is needed.

The manuscript requires significant language editing for clarity and scientific tone. Several sections are repetitive or loosely organized.

Moreover, the manuscript lacks a clear and concise conclusion that synthesizes the key findings and their biological or clinical implications. A strong closing section is needed to articulate the overall contribution of the study.

Minor Issues

GWAS results need QQplot.

The format of table is not consisted.

Limitations should be stated more explicitly

Reviewer #4: In this paper, the authors systematically identified key genes and potential drug targets associated with colorectal cancer in East Asian populations by integrating GWAS and RNA-Seq data, combined with molecular docking techniques. The authors also performed differential expression analysis on RNA-Seq data from TCGA to refine the findings. Overall, the paper is well-written. Below are my comments on the paper.

• In 1.2 Data Acquisition, the author mentioned that “Datasets were filtered based on case-control status and sample size, with smaller sample sizes excluded to ensure statistical robustness.” Could the author be more specific on how they determine if the sample size of a dataset is small.

• It would be better if the authors could briefly introduce the three GWAS dataset used. Are there any population substructure in each of the dataset?

• It seems to me that the authors are more interested in identifying CRC-related genes. If this is the case, what about using set-based genetic association analysis (e.g. SKAT by Wu et al. (2011)) directly to identifying disease-related genes? Are there any advantages of using SNP-based GWAS compared to set-based GWAS in your study?

Ref: Wu MC, Lee S, Cai T, Li Y, Boehnke M, Lin X. Rare-variant association testing for sequencing data with the sequence kernel association test. Am J Hum Genet. 2011 Jul 15;89(1):82-93. doi: 10.1016/j.ajhg.2011.05.029. Epub 2011 Jul 7. PMID: 21737059; PMCID: PMC3135811.

Reviewer #5: General Comments:

This study attempts an integrative approach combining GWAS and RNA-Seq data with molecular docking to identify therapeutic targets and drug candidates for colorectal cancer. The authors aim to bridge genetic insights with drug discovery, which is a valuable objective. However, the current methodology and presentation have several limitations that significantly impact the robustness and reliability of the findings. The conclusions drawn appear to be overstated given the foundational weaknesses.

Specific Comments:

1. The study uses three separate GWAS datasets. Given this, a formal meta-analysis of the GWAS data would significantly increase statistical power and the robustness of identified SNPs and genes. Alternatively, only retaining consistently replicated significant findings across at least two datasets could enhance accuracy and reduce false positives.

2. The significance threshold for SNP selection (p<1e−5) is relatively loose for GWAS, especially without stringent meta-analysis or replication across multiple independent cohorts. This increases the risk of including false-positive associations. Please justify this threshold more thoroughly or consider a stricter criterion.

3. The inclusion criteria for GWAS datasets mentions filtering based on "smaller sample sizes excluded to ensure statistical robustness". However, specific thresholds for exclusion are not provided. Please clarify what constitutes a "smaller sample size" in this context.

4. The linkage disequilibrium (LD) filtering criteria (r2<0.001 and a window size of 10,000 kb) seem unusually broad, potentially leading to very few independent SNPs or a lack of fine-mapping resolution. Please provide a more detailed justification for these specific parameters and their implications for instrumental variable selection.

5. The methods state that "genes and samples with more than 50% missing (NA) values were removed." This is a very lenient threshold, and keeping data with nearly 50% missingness, especially when followed by imputation, raises concerns about data quality and potential biases introduced by the imputation process. Please justify why such a high percentage of missing values was tolerated and discuss its potential impact.

6. The fold-change threshold of 2.0 (equivalent to |logFC|>1.0) for differential expression is not quite strict. Consider if the current stringency is appropriate for the downstream analyses.

7. Colorectal cancer incidence and mortality rates are noted to differ between males and females. A sex-stratified analysis could reveal important biological differences and sex-specific therapeutic targets. Please consider performing such an analysis or discussing its implications as a limitation and future direction.

8. The results show very few genes being shared across the three GWAS datasets. This lack of strong replication across datasets is a concern. The authors should explicitly discuss potential reasons for this limited overlap (e.g., population-specific effects, heterogeneity in study design, statistical power differences, or lack of truly robust signals) and its implications for the generalizability of the findings.

9. The GO and KEGG enrichment analyses appear weak, with many terms including only one or two genes. This makes it difficult to draw strong biological conclusions. The background gene set used for the enrichment analysis should be clearly indicated. Were all genes in the human genome used as background, or a more relevant subset (e.g., all genes expressed in colon/rectal tissue)? The adjusted p-values for many terms in Table 3 and Table 4 are relatively high, suggesting marginal significance despite being "top 10." The conclusions drawn from these enrichment results should be very carefully tempered.

10. Please clarify why adjusted p-values were not used for the online drug screening results. Unadjusted p-values in such a large-scale screening can lead to a high number of false positives.

11. The gene PYGL is highlighted as a key finding, but no validation is presented using external data or wet-lab experiments.

12. The most significant limitation of this study is the complete lack of experimental validation for the identified therapeutic targets and the candidate small-molecule inhibitors. The findings are entirely computational, and without in vitro or in vivo validation, their therapeutic potential remains speculative. This should be explicitly stated as a major limitation in the discussion, with a clear commitment to future experimental work. The authors mention "the incorporation of extensive positive and control datasets enhances the robustness and reliability of the findings" in the discussion of limitations. While this is a good practice for computational studies, it does not substitute for independent experimental validation. This statement should be rephrased to avoid overstating the current study's robustness in a biological context.

13. The supplementary table file names do not consistently match the table numbers inside the files.

Reviewer #6: To the Author,

Thanks for this valuable work.

Its noteworthy, this article, may lead to applied the suggested targeted therapies in experimental or clinical trial after subjected it to validation as well as further laboratory evaluations and adjustments. Furthermore, other limitation may the restriction of the study to the Asian population, which may not align with the genetic predispositions in other populations.

**Do you want your identity to be public for this peer review?** For information about this choice, including consent withdrawal, please see our Privacy Policy

Reviewer #1: No

Reviewer #2: No

Reviewer #3: No

Reviewer #4: No

Reviewer #5: No

Reviewer #6: **Yes: ** Luma Hassan Alwan Al Obaidy

---

## [Author Response · Author response to Decision Letter 1]

11 Jul 2025

Reviewer #5 provided several insightful comments on the manuscript, all of which were highly professional and offered valuable guidance for revision. These suggestions significantly improved the logical flow and academic rigor of the paper. We sincerely appreciate Reviewer #5’s constructive feedback

---

## [Decision Letter · Decision Letter 1]

6 Aug 2025

Dear Dr. liu,

Thank you for submitting your manuscript to PLOS ONE. After careful consideration, we feel that it has merit but does not fully meet PLOS ONE’s publication criteria as it currently stands. Therefore, we invite you to submit a revised version of the manuscript that addresses the points raised during the review process.

We look forward to receiving your revised manuscript.

Kind regards,

Zhengrui Li

Academic Editor

PLOS ONE

Journal Requirements:

Reviewers' comments:

Reviewer's Responses to Questions

**Comments to the Author**

Reviewer #1: (No Response)

Reviewer #3: All comments have been addressed

Reviewer #4: All comments have been addressed

Reviewer #5: (No Response)

Reviewer #6: All comments have been addressed

2. Is the manuscript technically sound, and do the data support the conclusions?

Reviewer #1: Partly

Reviewer #3: Yes

Reviewer #4: Yes

Reviewer #5: No

Reviewer #6: Yes

3. Has the statistical analysis been performed appropriately and rigorously?

Reviewer #1: No

Reviewer #3: Yes

Reviewer #4: Yes

Reviewer #5: No

Reviewer #6: Yes

4. Have the authors made all data underlying the findings in their manuscript fully available?

Reviewer #1: Yes

Reviewer #3: Yes

Reviewer #4: Yes

Reviewer #5: Yes

Reviewer #6: Yes

5. Is the manuscript presented in an intelligible fashion and written in standard English?

Reviewer #1: Yes

Reviewer #3: Yes

Reviewer #4: Yes

Reviewer #5: Yes

Reviewer #6: Yes

Reviewer #1: The prior comments still hold true regarding the lack of novelty in data or methodology for the analysis and results presented in the study.

Reviewer #3: This manuscript presents a promising integrative in silico approach combining GWAS, RNA-Seq, and molecular docking to identify therapeutic targets for colorectal cancer. The authors have addressed many of the reviewers' previous concerns, including data selection, method transparency, and overstatement of claims. However, several key issues remain unresolved:

The lack of direct SNP-expression linkage (e.g., eQTL or TWAS) weakens the connection between genetic association and expression.

nsufficient explanation for selecting PYGL for molecular docking over other candidates (e.g., SMAD7, TCF7L2).

Paper foamting and Language still to be polished.

Still want have QQ plot to validate SNP associations.

Reviewer #4: The authors have successfully addressed all my previous comments and I don't have additional comments.

Reviewer #5: General Comments:

The authors of this manuscript present a computational workflow to identify potential therapeutic targets and drug candidates for colorectal cancer (CRC) by integrating data from a Genome-Wide Association Study (GWAS) with RNA-sequencing (RNA-Seq) data. The study culminates in a virtual drug screening of a selected target, PYGL. While the overall objective of integrating multi-omics data for drug discovery is a valuable endeavor, the current manuscript suffers from fundamental methodological weaknesses, lack of transparency, and a disconnect between the claims and the supporting evidence. The revisions made in response to previous reviewer comments do not adequately address these core issues, and in some cases, introduce new inconsistencies. For these reasons, the manuscript is not suitable for publication in its current form.

Specific Comments:

1. The manuscript initially mentions using three GWAS datasets but the revised version states that only one dataset, GCST90018808, was used. This change, supposedly in response to requests for replication and meta-analysis, is not a sound scientific solution. Removing datasets without clear, predefined criteria and without acknowledging the loss of statistical power and generalizability is problematic. The authors' response to Reviewers simply states that the single dataset was selected without explaining the rationale for discarding the others. A robust study would perform a meta-analysis to combine evidence from multiple cohorts or, at a minimum, demonstrate consistent findings (replication) across them. The arbitrary selection of a single dataset makes the findings highly susceptible to a single study's biases and reduces the ability to generalize results.

The authors claim to have filtered datasets based on sample size but only provide a non-specific cutoff of ncase < 5,000 as an example. The specific criteria used to select GCST90018808 over the others and what constituted a "small sample size" for the excluded datasets remain unclear in the text. This lack of transparency makes the data selection process appear arbitrary.

2. The authors mention the study population for the GWAS data but fail to provide the same demographic information for the RNA-Seq data, creating a data-context gap. This information is essential for interpreting the results and discussing generalizability.

A central claim of the study is the integration of GWAS and RNA-Seq data. However, the manuscript fails to explicitly address whether the populations in the GWAS and RNA-Seq datasets are matched. The GWAS data is from European and East Asian populations , while the RNA-Seq data is from the TCGA-COAD cohort. The TCGA is a US-based consortium, and its racial and ethnic composition may not align with the East Asian population that the authors repeatedly emphasize as a focus in their introduction and discussion. A failure to match populations can introduce significant confounding due to population stratification, undermining the validity of linking genetic variants to gene expression.

The authors mention using the TwoSampleMR R package and filtering for weak instrumental variables, but they explicitly state that they did not perform a Mendelian Randomization (MR) analysis. Furthermore, they fail to perform an eQTL analysis to directly link the identified SNPs to gene expression changes, which is a critical missing piece of the claimed "integrative analysis". Without this crucial link, the association between the selected genes and CRC is not robustly established, and the integration of GWAS and RNA-Seq data remains superficial and unconvincing.

3. The manuscript contains contradictory statements regarding data preprocessing. For RNA-Seq data, the method mentions filtering out genes and samples with more than 1% missing (NA) values, but the results section mentions excluding genes with more than 50% missing values. This is a significant discrepancy that raises serious questions about the rigor and reproducibility of the methodology.

The Abstract is poorly structured. It mixes methodological steps with results, making it difficult to follow the logical flow of the study. For instance, it lists specific numbers of identified mRNAs and lncRNAs in the "Method" section, which should be in the "Result" section. Similarly, the Conclusion section provides new examples of genes (CDKN2B, BOC, and METRNL) that are listed as potential therapeutic targets, which should be presented and discussed in the main result.

The manuscript lacks a concise and impactful conclusion. The concluding paragraphs are repetitive and contain general statements about the value of computational methods, rather than synthesizing the specific and unique contributions of this study.

4. The authors acknowledge the lack of experimental validation but their responses and the manuscript's tone do not sufficiently temper the "overstated claims" of identifying "promising therapeutic targets". The claim that "the incorporation of extensive positive and control datasets enhances the robustness and reliability of the findings" is misleading, as this does not substitute for independent biological validation and is not adequately supported by the current work.

While survival analysis is a valuable component of the study, the authors do not specify if they performed a multiple testing correction for the survival analysis of the 18 screened genes. Given the number of tests performed (at least 18 genes x 3 survival metrics = 54 tests), uncorrected p-values (p < 0.05) are highly susceptible to false positives. This should be clearly stated and, if not performed, acknowledged as a limitation.

The GO and KEGG enrichment analyses are based on a very small gene list and, as the authors acknowledge in their response to Reviewer, often include only one or two genes per term. This provides minimal insight and weakens the biological interpretation of the findings. The conclusion that these analyses are meaningful and support the findings is not well-justified.

5. The GWAS Manhattan plot (Fig 2) shows a significance threshold of p<1e−6 , but the text also mentions a genome-wide significance threshold of p>5e−8, which is typically a much stricter standard. This inconsistency needs clarification.

Reviewer #6: The study attempts to find potent therapeutic targets in colorectal cancer by analyzing the genomic data of European and East Asian populations. The researchers suggested the PYGL gene as a potential therapeutic target, as it has interacting motives with the suggested therapeutic small molecules. All work was performed by a computation prediction application, which may facilitate finding targets and therapies for cancer and reduce the time of experimentation. The major limitation of this work is the lack of validation and experimentation, as the researcher confesses in their work.

**Do you want your identity to be public for this peer review?** For information about this choice, including consent withdrawal, please see our Privacy Policy

Reviewer #1: No

Reviewer #3: No

Reviewer #4: No

Reviewer #5: No

Reviewer #6: No

---

## [Author Response · Author response to Decision Letter 2]

15 Aug 2025

Reviewer #1: The prior comments still hold true regarding the lack of novelty in data or methodology for the analysis and results presented in the study.

Response: Thank you for your time and expertise.

Reviewer #3: This manuscript presents a promising integrative in silico approach combining GWAS, RNA-Seq, and molecular docking to identify therapeutic targets for colorectal cancer. The authors have addressed many of the reviewers' previous concerns, including data selection, method transparency, and overstatement of claims. However, several key issues remain unresolved:

The lack of direct SNP-expression linkage (e.g., eQTL or TWAS) weakens the connection between genetic association and expression.

nsufficient explanation for selecting PYGL for molecular docking over other candidates (e.g., SMAD7, TCF7L2).

Paper foamting and Language still to be polished.

Still want have QQ plot to validate SNP associations.

Response: Thank you for your time and expertise. The manuscript formatting and language have been thoroughly edited. A QQ plot has been added to the Results section.

Section 1.8 outlines the gene screening methodology.

“Structural data were obtained from the AlphaFold Protein Structure Database and the Protein Data Bank (PDB), with priority given to PDB entries. Selection criteria included the scientific name of the source organism (Homo sapiens), experimental method (X-ray diffraction or electron microscopy), and a refinement resolution of ≤2.5 Å.

A total of 17 mRNAs were identified through screening, among which only 3 possessed PDB IDs and contained small-molecule ligands. Based on the properties and size of the active sites, PYGL was ultimately selected for further investigation.

Table 1. Structural data of the 17 mRNA-encoded proteins

No. Gene symbol RNA type PDB AlphaFold

PDB ID Ligand ID

1 CDKN2B mRNA P42772

2 BOC mRNA 3N1G /

3 SCG5 mRNA P05408

4 LRP1 mRNA 1D2L /

5 ZCWPW2 mRNA 4Z0R EDO: C2H6O2

6 PYGL mRNA 3DDS 26B: C29H35N3O5

7 PREX1 mRNA 6VSK /

8 SMAD7 mRNA 7CD1 /

9 SLCO2A1 mRNA Q92959

10 METRNL mRNA Q641Q3

11 POU5F1B mRNA Q06416

12 ABHD12B mRNA Q7Z5M8

13 TCF7L2 mRNA 1JDH /

14 CDH3 mRNA 5JYL GOL: C3H8O3

15 SOX6 mRNA P35712

16 NKX2-3 mRNA Q8TAU0

17 TMEM238L mRNA A6NJY4

Reviewer #4: The authors have successfully addressed all my previous comments and I don't have additional comments.

Response: Thank you for your time and expertise.

Reviewer #5: General Comments:

1. The manuscript initially mentions using three GWAS datasets but the revised version states that only one dataset, GCST90018808, was used. This change, supposedly in response to requests for replication and meta-analysis, is not a sound scientific solution. Removing datasets without clear, predefined criteria and without acknowledging the loss of statistical power and generalizability is problematic. The authors' response to Reviewers simply states that the single dataset was selected without explaining the rationale for discarding the others. A robust study would perform a meta-analysis to combine evidence from multiple cohorts or, at a minimum, demonstrate consistent findings (replication) across them. The arbitrary selection of a single dataset makes the findings highly susceptible to a single study's biases and reduces the ability to generalize results.

The authors claim to have filtered datasets based on sample size but only provide a non-specific cutoff of ncase < 5,000 as an example. The specific criteria used to select GCST90018808 over the others and what constituted a "small sample size" for the excluded datasets remain unclear in the text. This lack of transparency makes the data selection process appear arbitrary.

Response: Thank you for your time and expertise.

2. The authors mention the study population for the GWAS data but fail to provide the same demographic information for the RNA-Seq data, creating a data-context gap. This information is essential for interpreting the results and discussing generalizability.

A central claim of the study is the integration of GWAS and RNA-Seq data. However, the manuscript fails to explicitly address whether the populations in the GWAS and RNA-Seq datasets are matched. The GWAS data is from European and East Asian populations , while the RNA-Seq data is from the TCGA-COAD cohort. The TCGA is a US-based consortium, and its racial and ethnic composition may not align with the East Asian population that the authors repeatedly emphasize as a focus in their introduction and discussion. A failure to match populations can introduce significant confounding due to population stratification, undermining the validity of linking genetic variants to gene expression.

The authors mention using the TwoSampleMR R package and filtering for weak instrumental variables, but they explicitly state that they did not perform a Mendelian Randomization (MR) analysis. Furthermore, they fail to perform an eQTL analysis to directly link the identified SNPs to gene expression changes, which is a critical missing piece of the claimed "integrative analysis". Without this crucial link, the association between the selected genes and CRC is not robustly established, and the integration of GWAS and RNA-Seq data remains superficial and unconvincing.

Response: Thank you for your time and expertise.

3. The manuscript contains contradictory statements regarding data preprocessing. For RNA-Seq data, the method mentions filtering out genes and samples with more than 1% missing (NA) values, but the results section mentions excluding genes with more than 50% missing values. This is a significant discrepancy that raises serious questions about the rigor and reproducibility of the methodology.

The Abstract is poorly structured. It mixes methodological steps with results, making it difficult to follow the logical flow of the study. For instance, it lists specific numbers of identified mRNAs and lncRNAs in the "Method" section, which should be in the "Result" section. Similarly, the Conclusion section provides new examples of genes (CDKN2B, BOC, and METRNL) that are listed as potential therapeutic targets, which should be presented and discussed in the main result.

The manuscript lacks a concise and impactful conclusion. The concluding paragraphs are repetitive and contain general statements about the value of computational methods, rather than synthesizing the specific and unique contributions of this study.

Response: Thank you for your time and expertise.

1% missing (NA) values The missing value threshold stated as 1% in the original text should be corrected to 50%, which was applied as a filtering criterion during data normalization using log2(X+1) transformation.

50% missing values For the Limma analysis, samples with zero expression values in >50% of genes were removed.

Both the Abstract and Conclusion sections have been revised.

4. The authors acknowledge the lack of experimental validation but their responses and the manuscript's tone do not sufficiently temper the "overstated claims" of identifying "promising therapeutic targets". The claim that "the incorporation of extensive positive and control datasets enhances the robustness and reliability of the findings" is misleading, as this does not substitute for independent biological validation and is not adequately supported by the current work.

While survival analysis is a valuable component of the study, the authors do not specify if they performed a multiple testing correction for the survival analysis of the 18 screened genes. Given the number of tests performed (at least 18 genes x 3 survival metrics = 54 tests), uncorrected p-values (p < 0.05) are highly susceptible to false positives. This should be clearly stated and, if not performed, acknowledged as a limitation.

The GO and KEGG enrichment analyses are based on a very small gene list and, as the authors acknowledge in their response to Reviewer, often include only one or two genes per term. This provides minimal insight and weakens the biological interpretation of the findings. The conclusion that these analyses are meaningful and support the findings is not well-justified.

Response: The specified content has been deleted as requested.“the incorporation of extensive positive and control datasets enhances the robustness and reliability of the findings”

The content has been added in Section 1.5.“The generated p value does not include correction for multiple hypothesis testing,”

5. The GWAS Manhattan plot (Fig 2) shows a significance threshold of p<1e-6 , but the text also mentions a genome-wide significance threshold of p>5e−8, which is typically a much stricter standard. This inconsistency needs clarification.

Response: While the conventional genome-wide significance threshold is typically set at p < 5e-8, we considered single nucleotide polymorphisms (SNPs) with p < 1e-6 as potentially significant in our study. Using this less stringent threshold (p <1e-6) for initial screening, we identified and validated several SNPs. Subsequent analysis revealed three genes CDKN2B, BOC, and METRNL showing significant associations, with p-values ranging between 5e-8 and 1e-6.

Reviewer #6: The study attempts to find potent therapeutic targets in colorectal cancer by analyzing the genomic data of European and East Asian populations. The researchers suggested the PYGL gene as a potential therapeutic target, as it has interacting motives with the suggested therapeutic small molecules. All work was performed by a computation prediction application, which may facilitate finding targets and therapies for cancer and reduce the time of experimentation. The major limitation of this work is the lack of validation and experimentation, as the researcher confesses in their work.

Response: Thank you for your time and expertise.

---

## [Decision Letter · Decision Letter 2]

10 Sep 2025

Integrative GWAS and RNA-Seq Analysis for Target Identification and Virtual Drug Screening in Colorectal Cancer

PONE-D-25-19455R2

Dear Dr. liu,

We’re pleased to inform you that your manuscript has been judged scientifically suitable for publication and will be formally accepted for publication once it meets all outstanding technical requirements.

Kind regards,

Zhengrui Li

Academic Editor

PLOS ONE

Additional Editor Comments (optional):

Reviewer #3:

Reviewers' comments:

Reviewer's Responses to Questions

**Comments to the Author**

Reviewer #3: (No Response)

2. Is the manuscript technically sound, and do the data support the conclusions?

Reviewer #3: Partly

3. Has the statistical analysis been performed appropriately and rigorously?

Reviewer #3: Yes

4. Have the authors made all data underlying the findings in their manuscript fully available?

Reviewer #3: Yes

5. Is the manuscript presented in an intelligible fashion and written in standard English?

Reviewer #3: Yes

Reviewer #3: The authors have made some revisions in response to the reviewers’ comments; however, many important points remain insufficiently addressed. In particular, the QQ plot raises concerns about potential inflation in the GWAS results, which should have been discussed in greater depth. Overall, the manuscript does not contain major fatal flaws, but it also lacks clear novelty or significant contributions that would advance the field.

**Do you want your identity to be public for this peer review?** For information about this choice, including consent withdrawal, please see our Privacy Policy

Reviewer #3: No

---

## [Editor Report · Acceptance letter]

PONE-D-25-19455R2

PLOS ONE

Dear Dr. liu,

I'm pleased to inform you that your manuscript has been deemed suitable for publication in PLOS ONE. Congratulations! Your manuscript is now being handed over to our production team.

Kind regards,

on behalf of

Dr. Zhengrui Li

Academic Editor

PLOS ONE